# Generalised Task Planning with First-Order Function Approximation

**Jun Hao Alvin Ng**
Edinburgh Centre for Robotics
Heriot-Watt University &
University of Edinburgh
Edinburgh, Scotland, United Kingdom
Alvin.Ng@hw.ac.uk

**Ronald P. A. Petrick**
Edinburgh Centre for Robotics
Heriot-Watt University
Edinburgh, Scotland, United Kingdom
R.Petrick@hw.ac.uk

**Abstract:** Real world robotics often operates in uncertain and dynamic environments where generalisation over different scenarios is of practical interest. In the absence of a model, value-based reinforcement learning can be used to learn a goal-directed policy. Typically, the interaction between robots and the objects in the environment exhibit a first-order structure. We introduce first-order, or relational, features to represent an approximation of the Q-function so that it can induce a generalised policy. Empirical results for a service robot domain show that our online relational reinforcement learning method is scalable to large scale problems and enables transfer learning between different problems and simulation environments with dissimilar transition dynamics.

**Keywords:** task planning, relational reinforcement learning, transfer learning

## 1 Introduction

Adaptation to different scenarios is essential for real world robotic applications. As a motivating example, consider a service robot interacting with people and objects as shown in Figure 1. The robot is required to attend to a person who requires assistance and complete the tasks given by the person. The uncertainty in who needs assistance, what these task could be, and the number of people and objects in the environment necessitates generalisation of learned knowledge from past observations in order to complete new but similar tasks in a different environment. If the model of the interaction between the robot and its environment is unknown due to its complexity, then value-based reinforcement learning (RL) methods can be used to solve the planning problems. The amount of observations required to achieve near-optimal behaviour, or sample complexity, is usually prohibitive, especially in large scale problems. This restricts the practicality of RL methods in applications where data collection is expensive. One solution is to utilise transfer learning which can reduce sample complexity by leveraging the knowledge learned in small scale problems where learning can be more efficient, or in simulated environments where data collection is cheap and safe.

Interactions between robots and objects in its environment often exhibit a first-order structure (i.e., ground actions of a lifted action change the relations or properties of all objects of the same type in similar manner). We model problems with a first-order structure using Relational Markov Decision Processes (RMDPs). Relational reinforcement learning (RRL) learns in a first-order representation rather than a propositional representation and can achieve knowledge transfer without the need of a mapping between problems [1]. In this work, we propose a model-free RRL method where the action-value function, or Q-function, is approximated by projecting the state space into a lower dimensional space using a set of features. We utilise an existing online feature discovery algorithm, iFDD+ [2], to incrementally add conjunctive state predicates as features to reduce the approximation error. Since state predicates are ground over objects, problems with different objects or numbers of objects have different sets of state predicates, or feature spaces, and generalisation is not possible. Instead, we use lifted state predicates, or first-order features, to approximate the Q-values of lifted actions rather than ground actions. By learning in a abstract state-action space which is common across all problems of the same domain (**related problems**), knowledge transfer is possible.

5th Conference on Robot Learning (CoRL 2021), London, UK.

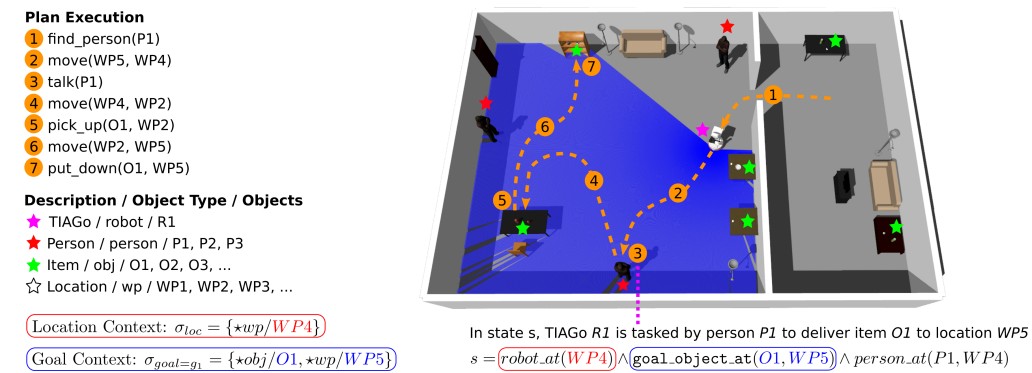

**Plan Execution**
1. find_person(P1)
2. move(WP5, WP4)
3. talk(P1)
4. move(WP4, WP2)
5. pick_up(O1, WP2)
6. move(WP2, WP5)
7. put_down(O1, WP5)

**Description / Object Type / Objects**
★ TIAGo / robot / R1
★ Person / person / P1, P2, P3
★ Item / obj / O1, O2, O3, ...
☆ Location / wp / WP1, WP2, WP3, ...

Location Context: $\sigma_{loc} = \{\star wp / WP4\}$
Goal Context: $\sigma_{goal=g_1} = \{\star obj / O1, \star wp / WP5\}$

In state s, TIAGo *R1* is tasked by person *P1* to deliver item *O1* to location *WP5*
$s = robot\_at(WP4) \wedge goal\_object\_at(O1, WP5) \wedge person\_at(P1, WP4)$

Figure 1: An environment in `Gazebo` for the `Service Robot` domain. A TIAGo robot fulfils a task from person $P1$ who requested assistance, by bringing an item $O1$ to a location $WP5$.

Our contribution is threefold. First, we propose a method to generate a first-order feature space automatically given a RMDP without the use of a model, expert knowledge, or training data. Second, we implement an online, model-free RRL method which learns a first-order linear function approximation of the Q-function given an initial set of first-order features. The approximated Q-function induces a generalised policy which allows transfer learning between related problems independent of the objects, number of objects, initial states, and goal states. Third, we introduce the concept of contextual knowledge to reduce granularity in the first-order function approximation, improve plan optimality, and reduce computational cost. We evaluate our method empirically on randomised problems in a service robot domain and in different simulation environments.

## 2 Related Work

Exploitation of the first-order structure of planning problems could reduce the sample complexity of RL methods. There are broadly two types of methods: model-based and model-free. Model-based methods such as [3, 4, 5, 6] learn relational models and employ planning techniques. Our work is a model-free RRL method and, therefore, we focus on reviewing model-free learning methods. Mausam and Weld [7] and Džeroski et al. [8] partition the state space into finer regions, each of which has a real-value representing the Q-value, using relational regression trees. Wu and Givan [9] perform supervised learning of relational features, represented as decision trees, by adding features which correlate well to the Bellman error of value functions. A decision tree is sensitive to the order of node splitting and is ill-suited in online RL where later observations yield new information which might necessitate the reconstruction of the trees. Ramon et al. [10] propose a tree restructuring operation but require statistics to be stored for every node. We use conjunctive features similar to `SVRRL` [11] which avoids the issue faced by decision trees. `SVRRL` represents a value function with a relational naive Bayes net and learns both the values and structure of the network. It utilises a distance metric to generalise over handcrafted non-binary relational features. Likewise, `RIB` [12], which uses instance-based learning where selected observed examples are stored, requires a domain-specific distance metric to compute Q-values of unseen state-action pairs. Our work does not require any distance metric to be defined. In these aspects, [13] and [14] are most similar to our work. The former samples features from a set of features with the use of prior training data while we discover features online. The latter restricts features to have at most one free variable while our features can have any number of free variables. Other works such as [8, 10, 15, 12] do not consider free variables. Instead, only objects in the arguments of the goal predicate and action are considered which restricts the applicability of their methods to problems with simple goals.

Some approaches require extensive domain knowledge to define specific representations that facilitate efficient learning. Morales [16] defines abstract actions (r-actions) and abstract states (r-states) in a relational representation, then learns policies in this abstract state-action space. Konidaris and Barto [17] learn a shaping reward in the action space, represented by sensory measurements which are common and hold the same semantics for all problems, to accelerate learning in another problem. In later work [18], options in the agent-space are learned and transferred. Van Otterlo [19] solves an abstract MDP, which is created with the use of background knowledge, of the underlying

---

**Algorithm 1:** RRL with Online Feature Discovery

---

**Input:** RMDP $(\mathcal{O}, \mathcal{P}, \mathcal{S}, \mathcal{A}, \mathcal{T}, \mathcal{R}, s_0, H, \gamma)$, Discovery threshold $\xi$

1  Initialise: $\boldsymbol{\Phi} \leftarrow$ initialise_features$(\mathcal{P}, \mathcal{A})$; $\boldsymbol{w} = \boldsymbol{0}$; $\widetilde{Q} := \boldsymbol{w}^T \boldsymbol{\Phi}$

2  **for** $t = 0$ **to** $H - 1$ **and** $s_t$ *is not terminal state* **do**

3  $\quad$ $a_t \leftarrow \pi_{\widetilde{Q}}(s_t)$

4  $\quad$ $s_{t+1}, r_t, \Delta_t \leftarrow$ Execute action $a_t$ in simulation environment or real-world

5  $\quad$ TD error $\delta_t = r_t + \gamma \max_{a \in \mathcal{A}} \widetilde{Q}(s_{t+1}, a) - \widetilde{Q}(s_t, a_t)$; Update $\boldsymbol{w}$

6  $\quad$ $\eta_t(\phi_i) = \dfrac{\left| \sum_{j=0, \phi_i(s_j, \hat{a}_j)=1}^{t} \delta_j \right|}{\sqrt{\sum_{j=0, \phi_i(s_j, \hat{a}_j)=1}^{t} 1}} \forall \phi_i \in \mathbb{P}(\boldsymbol{\Phi}_a)$; **if** $\eta_t(\phi_i) > \xi$ **then** append $\phi_i$ to $\boldsymbol{\Phi}_a$ and 0 to $\boldsymbol{w}$

---

RMDP. Sharma et al. [20] perform case-based learning and quantify state similarity by the Euclidean distance between a pair of states which are represented by handcoded, real-valued state variables. The Q-values of an action are a weighted sum of the Q-values of matching cases, weighted by how similar these cases are to the current state. Our work uses trivial domain knowledge for contextual grounding which we posit are known in most, if not all, planning problems anyway.

## 3 Preliminaries

We solve planning problems represented with RMDPs using RL methods where the Q-function is approximated with a linear function approximation.

**Relational Markov Decision Process (RMDP).** Factored Markov Decision Processes (MDPs) [21] represent a state $s$ with a set of state predicates $\mathcal{P}$ where $s = \bigwedge_{i=1}^{|\mathcal{P}|} p_i$ and $p_i \in \mathcal{P}$. We use symbols with boldface to indicate sets. If the transition of $p_i$ depends only on a small number of state predicates $\bar{\mathcal{P}} \subset \mathcal{P}$, then the transition function $\mathcal{T}$, which defines a probability distribution over possible successor states after executing an action, can be represented compactly; for a predicate $p_i$, $\mathcal{T}(p'_i | \mathcal{P}, a) = \mathcal{T}(p'_i | \bar{\mathcal{P}}, a)$ where $p'_i$ is the predicate at the next time step and $a$ is an action. A RMDP is a first-order representation of a factored MDP given by the tuple $(\mathcal{O}, \mathcal{P}, \mathcal{S}, \mathcal{A}, \mathcal{T}, \mathcal{R}, s_0, H, \gamma)$. $\mathcal{O}$ is a set of objects, each associated with a type, $\mathcal{P}$ is defined over $\mathcal{O}$, $\mathcal{S}$ is a set of all possible state specifications over $\mathcal{O}$ and $\mathcal{P}$, $\mathcal{A}$ is the set of all possible instantiated actions, $\mathcal{T} : \mathcal{S} \times \mathcal{A} \times \mathcal{S} \rightarrow [0,1]$, $\mathcal{R} : \mathcal{S} \times \mathcal{A} \rightarrow \mathbb{R}$ is the reward function, $s_0$ is the initial state, $H$ is the planning horizon, and $\gamma$ is the discount factor. Each predicate is applied over a type-consistent tuple of objects. Different planning problems of a domain can be constructed where their RMDPs have different $\mathcal{O}$, initial states, and goal states. A **generalised policy** is a policy $\pi : s \rightarrow a$ which directly solves any problem of a domain.

**Linear Function Approximation.** The Q-function estimates the expected return, or Q-value $Q(s, a)$, of executing an action $a$ in the state $s$ and following a policy thereafter. The Q-function can be represented with some form of parametric function approximation such as a linear combination of features: $\widetilde{Q} = \sum_{i=1}^{k} w_i \phi_i$, where $\boldsymbol{\Phi} = \{\phi_1, \ldots, \phi_k\}$ is a set of features and $\boldsymbol{w} = \{w_1, \ldots, w_k\}$ is a set of scalar weights. A feature $\phi_i$ maps $(s, a)$ to a scalar. The learning objective is to find the set of features and weights such that $\widetilde{Q}$ closely approximates the optimal Q-function. A policy $\pi$ induced by the optimal Q-function maximises the total discounted reward $\sum_{t=0}^{H} \gamma^t r_t$, where $r_t$ is the reward observed at time step $t$. A feature partitions the state space into regions such that states in a region are considered the same for the purpose of approximating the Q-values. The coverage of a feature is the region of the state space for which the feature evaluates to non-zero [22]. Features with low coverage give fine granularity in approximation and thus have better accuracy than features with high coverage. This impacts the soundness of policies. Conversely, features with high coverage offer better generalisation as learning is done over a smaller number of partitions of the state space.

## 4 Online Learning with First-Order Linear Function Approximation

Our objective is to learn a generalised policy which can be used to directly solve every problem of the same domain. We present our online RRL method in Algorithm 1 which learns a linear

function approximation of the Q-function, $\widetilde{Q}$, where the features $\boldsymbol{\Phi}$ are first-order (initialised in line 1). Algorithm 1 uses `iFDD+` [2] for online feature discovery. $\widetilde{Q}$ is defined over the abstract state space (represented by the first-order features) and lifted action space which are the same in all related problems. This allows transfer learning where $\widetilde{Q}$ is learned in a problem and transferred directly to solve another problem. Thus, the policy $\pi_{\widetilde{Q}}$ induced by $\widetilde{Q}$ is a generalised policy. At each time step $t$ (line 2), the robot executes an action $a_t$ given by $\pi_{\widetilde{Q}}$ (lines 3 and 4) and observes the successor state $s_{t+1}$, reward $r_t$, and duration of execution $\Delta_t$. The temporal difference (TD) error is computed and the weights $\boldsymbol{w}$ are updated accordingly (line 5) with a TD learning method. We used Double Q-learning [23] with replacing eligibility traces for each feature [24]. `iFDD+` adds new features, which are the conjunction of two existing features in $\boldsymbol{\Phi}$, to $\boldsymbol{\Phi}$ if their relevances $\eta_t$ exceed the discovery threshold $\xi$ (line 6). This reduces the approximation errors of $\widetilde{Q}$ by introducing a finer granularity in $\widetilde{Q}$. In the remainder of this section, we elaborate on Algorithm 1.

## 4.1 Generalisation with First-Order Features

A feature $\phi_i$ is a set of conjunctive state predicates or a single state predicate which evaluates to 1 if it is satisfied in a state; otherwise it evaluates to 0. We denote a function approximation which uses ground state predicates for features as a **ground approximation**. Let $\widetilde{Q}^1$ be a ground approximation for a RMDP with state-action space $\boldsymbol{\mathcal{S}}^1 \times \boldsymbol{\mathcal{A}}^1$, and $\phi_i$ be one of its features. Evaluating the value of $\phi_i = p_1 \wedge \ldots \wedge p_n \in \boldsymbol{\mathcal{S}}^1$ is straightforward. However, in a second RMDP (i.e., a different planning problem) with state-action space $\boldsymbol{\mathcal{S}}^2 \times \boldsymbol{\mathcal{A}}^2$, $\phi_i$ cannot be evaluated if any of its constituent state predicates $p_1, ..., p_n$ is not in $\boldsymbol{\mathcal{S}}^2$. Extending this further, $\widetilde{Q}^1(s, a)$ does not map to a real value if $s \in \{\boldsymbol{\mathcal{S}}^2 - \boldsymbol{\mathcal{S}}^1\}$ or $a \in \{\boldsymbol{\mathcal{A}}^2 - \boldsymbol{\mathcal{A}}^1\}$. Thus, $\widetilde{Q}^1$ cannot induce a policy for the second RMDP unless $\boldsymbol{\mathcal{S}}^1 = \boldsymbol{\mathcal{S}}^2$ and $\boldsymbol{\mathcal{A}}^1 = \boldsymbol{\mathcal{A}}^2$. However, the two RMDPs are trivially identical with possibly different initial states and goal states. We are interested in transfer learning between related problems with different state-action spaces. This motivates our use of a **first-order approximation** which is independent of $\boldsymbol{\mathcal{S}}$, $\boldsymbol{\mathcal{A}}$, and $\boldsymbol{\mathcal{O}}$. In a ground approximation, the features for each ground action $a$, $\boldsymbol{\Phi}_a$, are initialised as the set of every state predicate and their negation: $\boldsymbol{\Phi} := [\boldsymbol{\Phi}_{a_1}, ..., \boldsymbol{\Phi}_{a_m}]$, where $m$ is the number of ground actions. We use $\hat{\ }$ to denote a lifted predicate or a set of lifted predicates and lowercase (uppercase) letters to represent variables (objects). In a first-order approximation, we instantiate a set of first-order features for each lifted action $\hat{a}$ instead of every ground action. Otherwise, $\widetilde{Q}$ will not be independent of $\boldsymbol{\mathcal{O}}$ because ground actions are defined over $\boldsymbol{\mathcal{O}}$. $\hat{a}$ can be ground with the variable binding $\sigma_a = \{x/X, y/Y\}$ to a ground action $\mathtt{a}(X, Y)$. For every $\hat{a} \in \widehat{\boldsymbol{\mathcal{A}}}$, we initialise $\boldsymbol{\Phi}_{\hat{a}}$ (line 1 of Algorithm 1) with the following steps:

1. We denote the set of ground actions for $\hat{a}$ as $\boldsymbol{\mathcal{A}}_{\hat{a}}$. The initial set of ground features for every ground action $a \in \boldsymbol{\mathcal{A}}_{\hat{a}}$ are lifted in accordance with the binding $\sigma_a$ where objects in the arguments of $a$ are substituted with their corresponding **bound variables**.

2. $\boldsymbol{\Phi}_{\hat{a}}$ is the union of the features from step 1. The remaining objects are substituted with **free variables**, where $\star obj$ denotes a free variable of type $obj$.

This procedure will generate the same set of first-order features for every related problem. Since $\widehat{\boldsymbol{\mathcal{A}}}$ is also the same, our first-order approximation has the same representation in all related problems. A first-order feature must be fully grounded in order to evaluate its value in a state. In contrast to bound variables which are ground with objects in the arguments of actions, free variables can be ground to any object of the same type. We discuss the grounding of free variables in Section 4.3.

**Service robot domain.** We introduce a novel domain, `Service Robot` (`SR`), which is illustrated in a `Gazebo` environment in Figure 1. In `SR`, a TIAGo robot has to assist some people. The robot navigates from location $wp_1$ to $wp_2$ with the action $\mathtt{move}(wp_1, wp_2)$. A person can request assistance (represented by the state predicate $\mathtt{need\_assistance}(person)$) which is a probabilistic exogenous effect (i.e., it is not due to the robot's actions). The robot does not initially know where the people are located and has to find them $(\mathtt{find}(person))$. It then moves to the person and talks to them $(\mathtt{talk}(person))$. If the person has requested assistance, the robot would receive some tasks (or goals) of two possible types: (1) bring an item $obj$ to location $wp$ $(\mathtt{goal\_object\_at}(obj, wp))$, or (2) deliver $obj$ to $person$ $(\mathtt{goal\_object\_with}(obj, person))$.

The robot could pick up an item $\big(\texttt{pick\_up}(obj, wp)\big)$, put down an item $\big(\texttt{put\_down}(obj, wp)\big)$, give an item to a person $\big(\texttt{give}(obj, person)\big)$, or take an item from a person $\big(\texttt{take}(obj, person)\big)$.

**Example 1.** *In a ground approximation, we use the predicates* $\bar{\mathcal{P}} = \{\texttt{need\_assistance}(P_1),$ *..., $\texttt{need\_assistance}(P_m)\}$ as features (i.e., $|\bar{\mathcal{P}}|$ number of features), where $P_i$ are objects of type $person$. In a first-order approximation, these predicates are lifted to a first-order feature. For the lifted action $\texttt{talk}(person)$, $\phi_i = \texttt{need\_assistance}(person)$. The number of first-order features does not increase with the number of $person$ objects unlike the ground approximation. This demonstrates that the first-order approximation does not scale with the number of objects and is independent of $\mathcal{O}$. To evaluate $\phi_i$, it must be grounded. The bound variable $person$ is substituted according to the variable binding of $\texttt{talk}$ (e.g., for $\texttt{talk}(P3)$, $person$ is substituted with $P3$). One of the first-order features for $\texttt{move}(wp_1, wp_2)$ is $\phi_j = \texttt{need\_assistance}(\star person)$; because $\texttt{move}$ does not have an argument of type $person$, $P_i$ is substituted with a free variable $\star person$. $\phi_j$ could be grounded by substituting $\star person$ to any object of type $person$.*

### 4.2 Online Feature Discovery

In Section 4.1, we describe the initialisation of $\mathbf{\Phi}$ where each feature is a lifted state predicate. There could be large approximation errors in $\widetilde{Q}$ as each of these feature has a high coverage. Hence, adding conjunctive state predicates, which have lower coverage, to $\mathbf{\Phi}$ could reduce the approximation error. The problem of feature discovery is to determine which features to add. We utilise $\texttt{iFDD+}$, a model-free, online feature discovery algorithm (line 6 of Algorithm 1). A set of candidate features $\mathbf{\Phi}_a^c$ consists of features which are the conjunction of any two features (its **parent features**) in $\mathbf{\Phi}_a$. We denote the generation of $\mathbf{\Phi}_a^c$ from $\mathbf{\Phi}_a$ as $\mathbb{P}(\mathbf{\Phi}_a)$. $\texttt{iFDD+}$ iteratively adds candidate features to $\mathbf{\Phi}_a$ at each time step if their cumulative absolute approximation errors, or relevances, exceed the discovery threshold $\xi$ which is a user-defined parameter. The relevance of a candidate feature is updated at each time step by accumulating the TD error only if the candidate feature is active. A feature is active if it is true in the state and is not a parent feature of any active feature. The weight of a newly added feature is equals to the sum of the weights of its parents. When a feature $\phi_i$ is added to $\mathbf{\Phi}_a$, this in turn generates new candidate features with $\phi_i$ as one of its parent features.

### 4.3 Contextual Grounding of Free Variables

To evaluate the values of first-order features in a state, they must be fully ground. This is required in Algorithm 1 to determine the Q-values (lines 3 and 5) and the active features and candidate features (line 6). The grounding influences the step update of $\boldsymbol{w}$ (line 5) and feature discovery (line 6). Bound variables are substituted with objects of the same type in the arguments of a ground action (see Example 1). Features with at least one free variable are not fully ground. The set of possible substitutions for these features consists of every combination of objects for the free variables. The number of substitutions for a first-order feature $\phi_i$ and for $\mathbf{\Phi}_{\hat{a}}$ are:

$$|\boldsymbol{\sigma}_{\phi_i}| = \prod_{v \in \boldsymbol{v}} |v|, \text{ and } |\boldsymbol{\sigma}_{\mathbf{\Phi}_{\hat{a}}}| = \prod_{\phi_i \in \mathbf{\Phi}_{\hat{a}}} |\boldsymbol{\sigma}_{\phi_i}|, \tag{1}$$

respectively, where $\boldsymbol{\sigma}_{\phi_i}$ ($\boldsymbol{\sigma}_{\mathbf{\Phi}_{\hat{a}}}$) is the set of possible substitutions for $\phi_i$ ($\mathbf{\Phi}_{\hat{a}}$), $\boldsymbol{v}$ is the set of free variables in $\phi_i$ and $|v|$ is the number of objects of the same type as $v$. If we consider every substitution (i.e., to treat a free variable like an existential quantifier) where $\phi_i$ is true in $s$ if there exists a substitution in $\boldsymbol{\sigma}_{\phi_i}$ which makes it true, then there would be many such states (e.g., in Example 1, $\texttt{need\_assistance}(\star person)$ will be true if any person needs assistance). Thus, first-order approximation has a coarse granularity which can deteriorate performance. We propose two methods to resolve this. First, we use **contextual knowledge** to reduce the number of objects which a free variable can be grounded to (i.e., reduce $|v|$ in Equation 1). Any substitution which conflicts with the substitution due to contextual knowledge is removed from $\boldsymbol{\sigma}_{\mathbf{\Phi}_{\hat{a}}}$. Two substitutions conflict if they substitute a free variable with a different object. We introduce two forms of contextual knowledge which require trivial domain knowledge: goal and location. Second, after contextual knowledge is applied, we select $\boldsymbol{\sigma}_M \subseteq \boldsymbol{\sigma}_{\mathbf{\Phi}_{\hat{a}}}$ such that a metric $M$ is maximised: $\boldsymbol{\sigma}_M = \{\arg\max_{\sigma \in \boldsymbol{\sigma}_{\mathbf{\Phi}_{\hat{a}}}} M(\mathbf{\Phi}_{\hat{a}}^{\sigma})\}$, where $\mathbf{\Phi}_{\hat{a}}^{\sigma}$ is the grounding of $\mathbf{\Phi}_{\hat{a}}$ with $\sigma$. We define $M(\mathbf{\Phi}_{\hat{a}}^{\sigma}) = \sum_{\phi_i \in \mathbf{\Phi}_{\hat{a}}^{\sigma}} \phi_i(s, a)(|\phi_i|)^2$, where $|\phi_i|$ is the number of state predicates in $\phi_i$. This maximises the number of active, complex features. The choice of $M$ must be dependent on the state only to preserve the Markovian property of RMDPs.

**Goal Context.** The goals $\mathcal{G}$ in a planning problem are state predicates which can be trivially determined from the definition of terminal states which are commonly assumed to be known. The objects in $g \in \mathcal{G}$ are used for grounding free variables of the same type: $\sigma_{goal=g} = \{\star x_1/X_1, \ldots, \star x_n/X_n\}$ where $g = \mathtt{p}(X_1, \ldots, X_n)$. We consider only active goals: a goal is active if it is known and has not been achieved (e.g., in SR, a goal is known only after a person has given a task). We assume that achieved goals are inconsequential for decision making in the present and future. The Q-values after applying goal context represent the expected values of actions for achieving the goal. This is similar to the goal-associated Q-function in [25] but is not the same as additive rewards as defined in [26] where each goal is assumed to contribute uniformly and additively to the reward. $\phi_i = 1$ if there exists a $g$ such that $\sigma_{goal=g}$ results in $\phi_i$ being true; the value does not change even if more of such goals exist. A goal typically gives a contextual grounding which is in conflict with another goal. We propose two goal selection schemes: (1) GA uses every goal with the maximal metric $M$, and (2) G1 uses only one goal at a time. G1 selects the next goal in an ordering of active goals, $\bar{\mathcal{G}}$, when the currently selected goal is achieved. The order in $\bar{\mathcal{G}}$ could affect the optimality and soundness of the policy. We determine $\bar{\mathcal{G}}$ from the state trajectories observed in the previous episode. A partial order $g_i < g_j$ is added if $g_i$ is achieved before $g_j$. $\bar{\mathcal{G}}$ is randomly shuffled at the start of an episode such that every partial order is satisfied. If every goal is achieved, then the total order is known though it might not be optimal. However, due to exploration in RL, the order of goals achieved might not be the same as $\bar{\mathcal{G}}$. Thus, a more optimal $\bar{\mathcal{G}}$ could still be found.

**Example 2.** *In SR, the goals are represented by the state predicates* $\mathtt{goal\_object\_with}(obj, person)$ *and* $\mathtt{goal\_object\_at}(obj, wp)$. *Suppose that the goals are to deliver item $O1$ to location $WP5$ and item $O2$ to person $P1$. The former is denoted as $g_1 = \mathtt{goal\_object\_at}(O1, WP5)$ and the latter as $g_2 = \mathtt{goal\_object\_with}(O2, P1)$. The goal $g_1$ involves the objects $O1$ with type obj and $WP5$ with type wp, and the corresponding contextual grounding is $\sigma_{goal=g_1} = \{\star obj/O1, \star wp/WP5\}$. Likewise, $\sigma_{goal=g_2} = \{\star obj/O2, \star person/P1\}$.*

**Location Context.** Mobile robots move in an environment and can only interact with objects in their vicinity. Following this observation, location context substitutes the free variable with the current location of the robot: $\sigma_{loc} = \{\star wp/WP\}$ if the robot is at location $WP$ of type $wp$. Location context can be used with the goal context if there is no conflict (i.e., they substitute free variables of different types)—this queries if the robot is at the same location as the goal of interest. If there is a conflict between goal context and location context, then either one takes precedence. We let location context take precedence in our experiments.

**Example 3.** *In SR, the state predicate* $\mathtt{robot\_at}(wp)$ *represents the location of a robot. If the robot is at $WP4$, then the contextual grounding is $\sigma_{loc} = \{\star wp/WP4\}$. Following Example 2, we can apply $\sigma_{goal=g_2}$ and $\sigma_{loc}$ together without any conflict to give the contextual grounding $\sigma_{loc,goal=g_2} = \{\star wp/WP4, \star obj/O2, \star person/P1\}$. On the other hand, $\sigma_{goal=g_1}$ and $\sigma_{loc}$ are conflicting as they ground $\star wp$ to $WP5$ and to $WP4$, respectively. They can still be applied together by letting either one takes precedence (i.e., ignore the grounding of the other one). If location context takes precedence, then $\sigma_{loc,goal=g_1} = \{\star obj/O1, \star wp/WP4\}$.*

### 4.4 Granularity of First-Order Approximations

The lifted state-action space $\widehat{\mathcal{S}} \times \widehat{\mathcal{A}}$ is often much smaller than the ground state-action space since a lifted predicate $\mathtt{p}(x, y)$ can be ground in $|x||y|$ ways. Since $\widehat{\mathcal{S}}$, $\widehat{\mathcal{P}}$, and $\widehat{\mathcal{A}}$ are independent of $\mathcal{O}$, the lifted state-action space does not increase with the scale of the planning problem which makes first-order approximations scalable to large scale problems (see Example 1). However, the downside is a coarser granularity in approximation. The maximum number of features is $\sum_{a \in \mathcal{A}} |\Phi(a)| = 2^{2|\mathcal{P}|} \times |\mathcal{A}|$ in a ground approximation, and $\sum_{\hat{a} \in \widehat{\mathcal{A}}} |\Phi_{\hat{a}}| = 2^{2|\widehat{\mathcal{P}}|} \times |\widehat{\mathcal{A}}|$ in a first-order approximation. Since $|\widehat{\mathcal{P}}| \leq |\mathcal{P}|$ and $|\widehat{\mathcal{A}}| \leq |\mathcal{A}|$ (they are equal if there is one object for each type), a first-order approximation uses fewer features to partition the state space. Therefore, the size of the partitions (or granularity) must be larger than a ground approximation. Furthermore, free variables in a first-order feature increase its coverage because there only needs to exist a substitution among several possible substitutions to make the feature true. This deteriorates performance if there are plateaus in the induced policy. Plateaus are regions of the state space where there are multiple non-optimal actions with the maximal Q-value. The coarse granularity limits the type of problems a first-order approximation can be applied in. Because first-order approximation does not consider every ground state predicates, it is suited for planning in factored MDPs with independent goals.

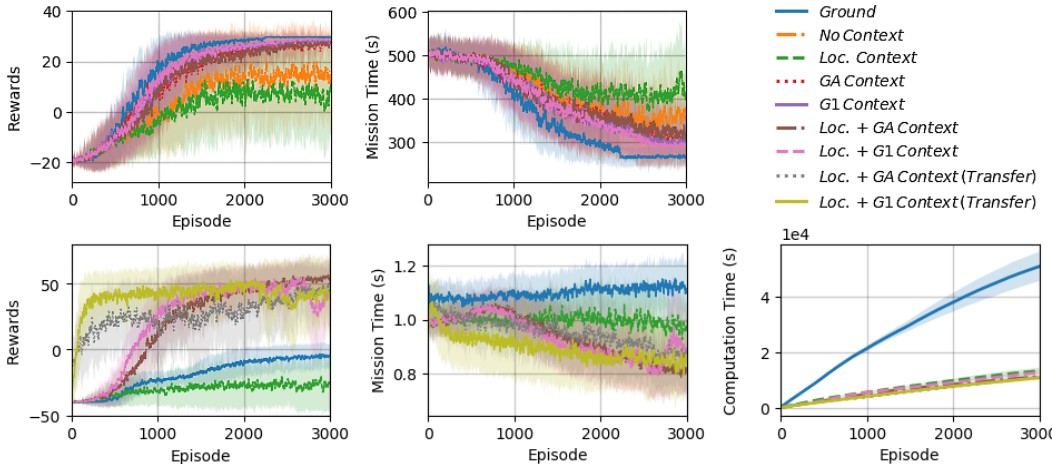

Figure 2: Results for $P_{small}$ (top row) and $P_{large}$ (bottom row) using RDDLSim as a simulator. The performance of first-order approximation with different combinations of contextual grounding is compared with ground approximation ($Ground$). Results for transfer learning ($Transfer$) from $P_{small}$ to $P_{large}$ are included as well. The shading represents one standard deviation.

|  | $P_{small}$ (ROS) | $P_{small}$ (RDDLSim) | $P_{large}$ (ROS) | $P_{large}$ (RDDLSim) |
|---|---|---|---|---|
| **Reward** | $22.7 \pm 10.6$ | $28.6 \pm 0.9$ | $45.2 \pm 21.0$ | $46.0 \pm 24.7$ |
| **Mission Time (s)** | $303.7 \pm 86.7$ | $289.8 \pm 30.3$ | $607.3 \pm 116.4$ | $648.6 \pm 122.3$ |

Table 1: Results for transfer learning from RDDLSim to ROS aggregated over 100 runs. This is compared with results from RDDLSim $\big($Loc.+G1 for $P_{small}$ and Loc.+G1 (Transfer) for $P_{large}\big)$.

## 5   Experiments and Results

We now demonstrate empirically that our work (1) learns a generalised and goal-directed policy, (2) enables transfer learning from small scale problems ($P_{small}$) to large scale problems ($P_{large}$) and between different simulation environments, and (3) reduces memory and computational costs. Results are averaged over 10 independent runs where a different, randomised problem is used. The problems are randomised in the locations of items and in the tasks given. We assume that the preconditions of actions are known (i.e., an illegal action will never be executed).

**Experimental Setup.** In $P_{small}$ ($P_{large}$), $\mathcal{O}$ contains three (six) items, one person (three people), and five (ten) locations; the probability of people needing assistance is 0.5 (0.5, 0.3, and 0.0 for three people). The size of the state-action spaces are $2^{56} \times 65$ for $P_{small}$ and $2^{216} \times 264$ for $P_{large}$. Each person who requires assistance gives two tasks when talked to. Executing an action provides a reward of $-1$ and completing a task (or goal) provides a reward of 20. We tested our work in two simulation environments: RDDLSim [27] and ROS (Robot Operating System) [28] with the ROSPlan [29, 30] package. Both simulators return an observation for executing an action in a state (line 4 of Algorithm 1). We augment the observation from RDDLSim with the execution duration $\Delta$ which is the mean of the observed $\Delta$ in ROS injected with normally distributed noise (standard deviation is 20% of the mean). We built a Gazebo environment (see Figure 1) for a more realistic simulation relative to RDDLSim. Firstly, in RDDLSim, the robot loses localisation when moving with a probability of 0.1; in ROS, it is deemed to have lost localisation when the covariance of the pose estimation from adaptive Monte-Carlo localisation exceeds a user-defined threshold. Secondly, in RDDLSim, the robot always succeeds in finding a person and ends up in the same location as the person. In ROS, the robot follows an exploration path to find a person. It might find the intended person and/or other people it encounters along the way, and might not be in the same location as the person since it could detect people from a distance. Lastly, in RDDLSim, picking up and putting down an item always succeeds while these actions might fail in ROS. Each experiment in RDDLSim is conducted on a single core AMD Opteron 6376 and with 8 GB of RAM.

**Ablation Study.** We conducted an ablation study for contextual grounding in first-order approximation and benchmarked it against ground approximation. RDDLSim was used. The results are shown in Figure 2. We measure performance by the total undiscounted reward received in each episode and by the total duration of plan execution (mission time). Executing an action incurs a reward of $-1$ which could result in a learned policy which minimises the number of actions executed; this seems to suffice in reducing the mission time as it was observed that the mission time reduces asymptotically while the reward increases. G1 and Loc.+G1 (combination of location context and G1) generally outperforms GA and Loc.+GA. This is because the goals in SR are independent of each other and considering one goal at a time reduces $\sigma_{\Phi_{\hat{a}}}$ and the coarseness of granularity more than GA does. In $P_{small}$, using no context or only location context has the worst performances. Due to our modelling choice to include $wp$ in arguments of actions $\big(\text{e.g., } \texttt{pick\_up}(obj, wp) \text{ and } \texttt{put\_down}(obj, wp)\big)$, features with $\star wp$ are unnecessary to approximate the Q-values of some actions—we opt against using domain knowledge to prune these features. Nevertheless, location context is crucial in reducing the memory and computational cost; experiments for $P_{large}$ without location context are computationally expensive and are omitted. The computation time for $P_{large}$ is further reduced when goal context is also used. The computation time for $P_{small}$ is comparable for all runs and is thus omitted. Although ground approximation has the best performance in $P_{small}$, it has the worst performance and the highest computational cost in $P_{large}$. The former is due to a lack of generalisation and the latter is due to the large set of features. For $P_{small}$, ground approximation has $6894 \pm 2.7$ (one standard deviation) features while the various configurations of first-order approximation have 280 to 316 features. In contrast, for $P_{large}$, ground approximation has $94.1 \times 10^3 \pm 8.1 \times 10^3$ features while the various configurations of first-order approximation have 382 to 417 features. Evidently, ground approximation has poorer scalability than first-order approximation.

**Transfer Learning.** In Figure 2, we show the results of transfer learning where $\widetilde{Q}$ trained in $P_{small}$ $\big(\widetilde{Q}^{P_{small}}\big)$ is used, and continues to be updated in $P_{large}$. The type of contextual grounding used in $P_{small}$ must also be used in $P_{large}$ for transfer learning to work. We consider only Loc.+G1 or Loc.+GA as these are computationally tractable. The former outperforms the latter due to the aforementioned reason. The boost in performance due to transfer learning is evident in episode 1. The asymptotic performance without transfer learning approaches that of transfer learning as expected. However, for Loc.+GA, transfer learning performed asymptotically poorer than without transfer learning. This is because GA considers multiple goals for contextual grounding. Thus, it generalises poorly from $P_{small}$, which has two goals, to $P_{large}$, which has four goals.

Lastly, we evaluate the generalisation from RDDLSim simulations to ROS simulations. We used ten randomised problems for $P_{small}$ and $P_{large}$ which are different from the ones used in RDDLSim. A greedy policy induced by $\widetilde{Q}^{P_{small}}$ using Loc.+G1 can directly solve $P_{small}$ but not $P_{large}$. Therefore, we trained $\widetilde{Q}^{P_{small}}$ further in 500 episodes of $P_{large}$ using RDDLSim with a modified transition function (i.e., the robot's location does not change when finding a person) to solve $P_{large}$. Without further training, the policy could fail to instruct the robot to move to a person after finding that person. The results are shown in Table 1 alongside the results from RDDLSim (i.e., Figure 2) for ease of comparison. For $P_{small}$, the performance in RDDLSim is better with smaller variances which is expected since the simulations do not consider real-world dynamics and are more optimistic. The results for $P_{large}$ are comparable for both simulation environments. Thus, the first-order approximations are successful at generalising over both simulation environments.

## 6   Conclusion

In this paper, we proposed a model-free, online RRL algorithm which learns an approximation of the Q-function with first-order features that is able to generalise across different problems of the same domain. Free variables in the first-order features caused ambiguity in how they should be substituted. The choice of a substitution influences the learning and is critical to performance. We resolve this problem by utilising contextual knowledge to substitute free variables. Empirical results showed that our method is scalable to large scale problems by keeping memory usage and computational cost tractable, and can generalise across problems of different scales and across different simulation environments. In future work, we plan to investigate other forms of contextual knowledge.

**Acknowledgments**

This work was partially funded by the EPSRC ORCA Hub (http://orcahub.org/) under grant number EP/R026173/1.

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

# A Appendix

## A.1 Relational Reinforcement Learning

We elaborate on our relational reinforcement learning method (an abridged version is outlined in Algorithm 1) which is shown in Algorithm 2. The inputs to the algorithm are a RMDP representing a planning problem, a set of features $\boldsymbol{\Phi}$ and weights $\boldsymbol{w}_0$ which approximate the Q-function $\widetilde{Q}$, candidate features $\boldsymbol{\Phi}^c$ with their relevance $\boldsymbol{\eta}$, the discovery threshold $\xi$ for iFDD+, learning rate $\alpha$, contextual knowledge $CK$ (if any), and a maximisation metric $M$. $\boldsymbol{\Phi}$, $\boldsymbol{w}_0$, $\boldsymbol{\Phi}^c$, and $\boldsymbol{\eta}$ are learned in another planning problem and transferred to the current problem. If transfer learning is not used, then they are the empty set $\varnothing$ and need to be initialised (lines 1 to 5). The initial set of first-order features $\boldsymbol{\Phi}$ is the concatenation of first-order features for each lifted action, $\boldsymbol{\Phi}_{\hat{a}}$. $\boldsymbol{\Phi}_{\hat{a}}$ is initialised with Algorithm 4 (line 4) which is described later. The weights are initialised to a vector of zeroes (line 5). In a linear function approximation, the Q-value of a state-action pair $(s, a)$ is the dot product of $\boldsymbol{w}$ and $\boldsymbol{\Phi}(s, a)$ (line 6). The latter is returned by Algorithm 3 which is described later.

Lines 7 to 25 show the planning cycle for $H$ time steps. To select an action using a policy $\pi_{\widetilde{Q}}$ generated by $\widetilde{Q}$ (line 8), the Q-values of every action must be evaluated. This implies that $\boldsymbol{\Phi}(s_t, a)$ must be evaluated for every $a \in \mathcal{A}$. Thus, Algorithm 3 is called $|\mathcal{A}|$ times. The action returned by $\pi_{\widetilde{Q}}$ is executed (line 10) which returns an observation $(s_{t+1}, r_t, \Delta_t)$ where $s_{t+1}$ is the successor state, $r_t$ is the reward, and $\Delta_t$ is the duration of the execution. The TD error is computed (line 12) and the weights are updated (line 13) following the observation.

The vector of real numbers, $\boldsymbol{\Phi}(s_t, a_t)$ (line 11), has a value of 1 if the corresponding feature is active. The set of active features is extracted (line 14). The set of active candidate features is the set of conjunction of every pair of active features in $\boldsymbol{\Phi}$ (line 15). For each active candidate feature, its relevance is computed (line 17). Candidate features with relevances greater than $\xi$ are added to $\boldsymbol{\Phi}$ with initial weights equal to the sum of the weights of their parent features (lines 18 to 22). New candidate features can be constructed from these newly added features in the next time step. The set of candidate features accumulated thus far, $\boldsymbol{\Phi}^c$, and its relevances, $\boldsymbol{\eta}$, are updated accordingly (lines 24 and 25). The algorithm terminates after $H$ time steps or if the terminal state is reached (line 7). It returns $\boldsymbol{w}$, $\boldsymbol{\Phi}$, $\boldsymbol{\Phi}^c$, and $\boldsymbol{\eta}$ (line 26) which are used as inputs for the next planning problem (i.e., transfer learning).

Algorithm 3 evaluates the binary value of each feature $\phi_f \in \boldsymbol{\Phi}$ and returns a vector of real numbers. The set of possible substitutions $\boldsymbol{\sigma}_{\boldsymbol{\Phi}}$ consists of every combination of objects for the free variables (line 2). We apply contextual grounding (line 3) by considering the state $s$ (e.g., to determine the goal(s) and the location of the agent). Then, a selection method selects a subset of substitutions from $\boldsymbol{\sigma}_{\boldsymbol{\Phi}} \cap \boldsymbol{\sigma}_{CK}$ (line 4) such that the metric $M$ is maximised. $\boldsymbol{\Phi}^{\sigma}$ represents the grounding of $\boldsymbol{\Phi}$ with $\sigma$. Lastly, an element-wise logical disjunction operator is applied to every grounding of $\boldsymbol{\Phi}$ with $\boldsymbol{\sigma}_M$ (line 5).

## A.2 Initialisation of First-Order Features

We describe the initialisation of a set of first-order features, $\boldsymbol{\Phi}_{\hat{a}}$, for each lifted action $\hat{a}$ as shown in Algorithm 4. The inputs are the set of state fluents $\mathcal{P}$ and the set of ground actions for $\hat{a}$, $\mathcal{A}_{\hat{a}}$. For each ground action $a \in \mathcal{A}_{\hat{a}}$, a set of ground features $\boldsymbol{\Phi}_a$ is initialised (`initialise_ground_features` in line 4). In this work, we used the set of every state fluent and their negation, as well as the non-fluents. Each feature $\phi_f \in \boldsymbol{\Phi}_a$ is lifted in accordance with the substitution of $a$, $\sigma_a$ (`lift` in line 5). A feature might be partially lifted if it contains objects that are not in the arguments of $a$. $\boldsymbol{\Phi}_{\hat{a}}$ is the union of these partially or fully lifted features for each $a \in \mathcal{A}_{\hat{a}}$ (line 5). Remaining objects in $\boldsymbol{\Phi}_{\hat{a}}$ are substituted with free variables to yield a set of fully lifted, or first-order, features (`quantify` in line 6).

## A.3 Domain Model

RDDL [27] is a planning language for describing planning problems, and is used in recent International Probabilistic Planning Competitions (IPPCs) [31, 32]. Semantically, RDDL describes dynamic Bayesian Networks extended with an influence diagram. A RDDL domain is described by object types, non-fluents, fluents, conditional probability functions (CPFs), and a reward func-

---
**Algorithm 2:** RRL with Online Feature Discovery (in details)
---

**Input:** RMDP $(\mathcal{O}, \mathcal{P}, \mathcal{S}, \mathcal{A}, \mathcal{T}, \mathcal{R}, s_0, H, \gamma)$,
        Features $\mathbf{\Phi}$,
        Weights $\boldsymbol{w}_0$,
        Candidate features $\mathbf{\Phi}^c$ with relevance $\boldsymbol{\eta}$,
        Discovery threshold $\xi$,
        Contextual knowledge $CK$,
        Maximisation metric $M$

1  **if** $\mathbf{\Phi} = \varnothing$ **then**
2     $\widehat{\mathcal{A}} \leftarrow$ Get lifted actions from $\mathcal{A}$
3     **for** $\hat{a} \in \widehat{\mathcal{A}}$ **do**
4          $\mathbf{\Phi} \leftarrow \mathbf{\Phi} \cup \text{initialise\_features}(\mathcal{P}, \mathcal{A}_{\hat{a}})$
5  **if** $\boldsymbol{w}_0 = \varnothing$ **then** $\boldsymbol{w}_0 \leftarrow \mathbf{0}$
6  $\widetilde{Q}(s,a) := \boldsymbol{w}^T \mathbf{\Phi}(s,a)$ where $\mathbf{\Phi}(s,a) \leftarrow \text{evaluate\_features}(\mathcal{O}, \mathbf{\Phi}, CK, M, s, a)$
7  **for** $t = 0$ **to** $H - 1$ **and** $s_t$ *is not terminal state* **do**
8     $a_t \leftarrow \pi_{\widetilde{Q}}(s_t)$
9     $\hat{a}_t \leftarrow \text{lift}(a_t)$
10    $s_{t+1}, r_t, \Delta_t \leftarrow$ Execute action $a_t$ in simulation environment or real-world
11    $\mathbf{\Phi}(s_t, a_t) \leftarrow \text{evaluate\_features}(\mathcal{O}, \mathbf{\Phi}, CK, M, s_t, a_t)$
12    $\delta_t = r_t + \gamma \max_{a \in \mathcal{A}} \widetilde{Q}(s_{t+1}, a) - \widetilde{Q}(s_t, a_t)$
13    Update $\boldsymbol{w}: w_{t+1}^i = w_t^i + \alpha \delta_t \frac{\phi_i(s,a)}{||\mathbf{\Phi}(s,a)||_1} \forall w_t^i \in \boldsymbol{w}$
14    $\mathbf{\Phi}_{active} \leftarrow$ Get features in $\mathbf{\Phi}$ which are active in $s$
15    $\mathbf{\Phi}_{active}^c = \mathbb{P}(\mathbf{\Phi}_{active})$
16    **for** $\phi_i \in \mathbf{\Phi}_{active}^c$ **do**
17        Relevance $\eta_t(\phi_i) = \dfrac{\left| \sum_{j=0, \phi_i(s_j, a_j)=1}^{t} \delta_j \right|}{\sqrt{\sum_{j=0, \phi_i(s_j, a_j)=1}^{t} 1}}$
18        **if** $\eta_t(\phi_i) > \xi$ **then**
19            Append $\phi_i$ to $\mathbf{\Phi}_{\hat{a}} \subset \mathbf{\Phi}$
20            Append $w_i = w_j + w_k$ to the sub-vector of $\boldsymbol{w}$ for $\hat{a}_t$ where $\phi_i = \phi_j \wedge \phi_k$
21            **if** $\phi_i \in \mathbf{\Phi}^c$ **then** remove $\phi_i$ from $\mathbf{\Phi}^c$
22            **if** $\eta_t(\phi_i) \in \boldsymbol{\eta}$ **then** remove $\eta_t(\phi_i)$ from $\boldsymbol{\eta}$
23        **else**
24            **if** $\phi_i \notin \mathbf{\Phi}^c$ **then** add $\phi_i$ to $\mathbf{\Phi}^c$
25            Add or update $\eta_t(\phi_i)$ in $\boldsymbol{\eta}$
26  **return** $\boldsymbol{w}, \mathbf{\Phi}, \mathbf{\Phi}^c, \boldsymbol{\eta}$

---
**Algorithm 3:** Evaluate a set of features
---

1  **Function** `evaluate_features`$(\mathcal{O}, \mathbf{\Phi}, CK, M, s, a)$:
    **Input:** Set of objects $\mathcal{O}$,
            Features $\mathbf{\Phi}$,
            Contextual knowledge $CK$,
            Maximisation metric $M$,
            State $s$,
            Ground action $a$
2    $\boldsymbol{\sigma}_{\mathbf{\Phi}} \leftarrow$ Get the set of possible substitutions from $\mathcal{O}$ for $\mathbf{\Phi}$
3    $\boldsymbol{\sigma}_{CK} \leftarrow$ Apply contextual grounding $CK$ given $s$
4    $\boldsymbol{\sigma}_M \leftarrow \left\{ \arg\max_{\sigma \in \boldsymbol{\sigma}_{\mathbf{\Phi}} \cap \boldsymbol{\sigma}_{CK}} M(\mathbf{\Phi}^{\sigma}(s, a)) \right\}$
5    $\mathbf{\Phi}(s, a) \leftarrow \bigvee_{\sigma \in \boldsymbol{\sigma}_M} \mathbf{\Phi}^{\sigma}(s, a)$
6    **return** $\mathbf{\Phi}(s, a)$

---

tion. State fluents (non-fluents) are state variables that change (do not change) with time. A RDDL problem is specified by objects, initial state, and values of non-fluents. A domain can have different problems by randomising the initial state and non-fluents. We have included the RDDL domain and problems files for our `Service Robot` (SR) planning problem as supplementary material. We briefly describe the state fluents, non-fluents, and actions for SR. We omitted the argument *robot* for brevity

---

**Algorithm 4:** Initialise a set of first-order features for a lifted action

---

1    **Function** initialise_features($\mathcal{P}, \mathcal{A}_{\hat{a}}$):
     **Input:** Set of state predicates $\mathcal{P}$,
              Set of ground actions $\mathcal{A}_{\hat{a}}$
2    $\Phi_{\hat{a}} = \varnothing$
3    **for** $a \in \mathcal{A}_{\hat{a}}$ **do**
4      $\Phi_a \leftarrow$ initialise_ground_features$(\mathcal{P}, a)$
5      $\Phi_{\hat{a}} \leftarrow \Phi_{\hat{a}} \cup \text{lift}\big(\Phi_a, \sigma_a\big)$
6    **return** quantify$\big(\Phi_{\hat{a}}\big)$

---

as SR is a single-agent problem but included them here for consistency with the RDDL domain and problem files.

A fluent or non-fluent is represented by its name and typed variables in its arguments. We use the type $robot$ to represent a robot, $wp$ to represent a location, $obj$ to represent an item, and $person$ to represent a person. The state fluents which describe the states and goals are:

- robot_at$(robot, wp)$: $robot$ is at location $wp$,
- localised$(robot)$: $robot$ is localised,
- emptyhand$(robot)$: $robot$ is not holding any items,
- holding$(robot, obj)$: $robot$ is holding item $obj$,
- object_at$(obj, wp)$: item $obj$ is at location $wp$,
- object_with$(obj, person)$: item $obj$ is with $person$,
- goal_object_at$(obj, wp)$: a goal is for item $obj$ to be at location $wp$,
- goal_object_with$(obj, person)$: a goal is for item $obj$ to be with $person$,
- person_at$(person, wp)$: $person$ is at location $wp$,
- need_assistance$(person)$: $person$ needs assistance,
- needed_assistance$(person)$: $person$ needed assistance (a $person$ only needs assistance once per problem), and
- reward_received$(obj)$: the goal involving $obj$ has been achieved (each $obj$ can be involved in at most one goal).

We use non-fluents to represent the location of each person and the tasks (goals) they would give. These non-fluents are not made known to Algorithm 2. The non-fluents are:

- PROB_NEED_ASSISTANCE$(person)$: the probability of $person$ needing assistance,
- PERSON_GOAL_OBJECT_AT$(person, obj, wp)$: $person$ will instruct the robot to place item $obj$ at location $wp$,
- PERSON_GOAL_OBJECT_WITH$(person_1, obj, person_2)$: $person_1$ will instruct the robot to bring item $obj$ to $person_2$ where $person_1$ and $person_2$ could refer to the same person, and
- PERSON_IS_AT$(person, wp)$: $person$ is at location $wp$ (this is different from person_at$(person, wp)$ which represents the knowedge of the robot).

The actions which the robot could execute are:

- move$(robot, waypoint1, waypoint2)$: $robot$ moves from $waypoint1$ to $waypoint2$,
- localise$(robot)$: $robot$ localises by performing simultaneous localisation and mapping (SLAM),
- find_person$(robot, person)$: $robot$ explores the room to find $person$ (this action terminates at the end of the exploration path or when $person$ is found),
- talk_to_person$(robot, person)$: $robot$ talks to $person$ $\big($if need_assistance$(person)$ is true, then two goals from $person$ will be made known and are represented by goal_object_at$(obj, wp)$ and/or goal_object_with$(obj, person)\big)$,

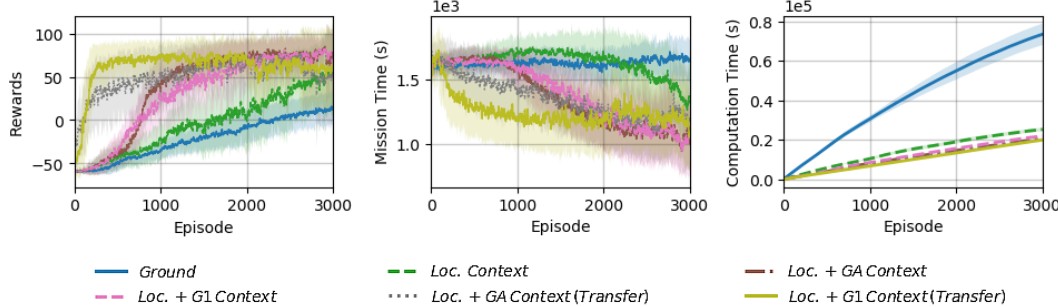

Figure 3: Results for an ablation study where different combinations of contextual grounding are used. The results are aggregated over 10 randomised large scale problems of `Service Robot` where three people need assistance (`SR3`). The shading represents one standard deviation. The simulator used is `RDDLSim`. For transfer learning ($Transfer$), the Q-function learned in `SR1` is transferred to `SR3`.

- `pick_up`($robot, obj, wp$): $robot$ picks up item $obj$ from location $wp$,
- `put_down`($robot, obj, wp$): $robot$ puts down item $obj$ at location $wp$,
- `take`($robot, obj, person$): $robot$ takes item $obj$ from $person$, and
- `give`($robot, obj, person$): $robot$ gives item $obj$ to $person$.

## A.4    Experiment Details

In our experiments, we used an $\epsilon$-$greedy$ policy with a linearly decaying $\epsilon$ over episodes. The parameters used are $\epsilon_{initial} = 1$ for experiments without transfer learning and $0.2$ otherwise, $\alpha = 0.3$ (learning rate), $\gamma = 0.9$, $\lambda = 0.7$ (decay rate for eligibility traces), and $\xi = 3$. $H$ is $20$ ($40$) for $P_{small}$ ($P_{large}$). We utilise two simulators in our experiments: `RDDLSim` [27] and `ROS` [28]. `RDDLSim` returns a successor state and reward based on the CPFs and reward function defined in the RDDL domain. It does not simulate a robot operating in an environment. For a more realistic simulation, we use `ROS` which considers aspects such as sensors (for localisation, navigation and detecting people), motion planning (for navigation, manipulation planning (for grasping), etc. For manipulation planning, we observed that the robot fails to grasp the item frequently. Since our work does not deal with manipulation planning, we simulated the success of grasping. If the robot fails to grasp an item, we would still consider it a success with a probability of $0.8$ and update the successor state accordingly (e.g., `holding`($robot, obj$) is true even if the robot fails to pick up an item $obj$ in `ROS`). Regardless, we observed that our learned policy will repeatedly attempt to pick up an item if it failed to in previous attempts.

## A.5    Additional Results for the Service Robot Domain

In Section 5, we present results for small scale (one $person$, three $obj$, and five $wp$) and large scale (three $person$, six $obj$, and 10 $wp$) problems. Here, we denote the former as `SR1` and the latter as `SR2`. In `SR1`, person $p1$ needs assistance with a probability of $0.5$. In `SR2`, person $p1$ needs assistance with a probability of $0.5$, person $p2$ needs assistance with a probability of $0.3$, and person $p3$ does not require assistance. We now consider a variant of `SR2` where $p3$ also needs assistance with a probability of $0.3$. We denote this variant as `SR3`. For `SR1`, `SR2`, and `SR3`, the problems are randomised in the initial locations of objects and the types of goals:

- `PERSON_GOAL_OBJECT_AT`($p1, obj, wp$): a goal from $p1$ where $obj$ and $wp$ are randomised object instances,
- `PERSON_GOAL_OBJECT_WITH`($p1, obj, p1$): a goal from $p1$ where $obj$ is a randomised object instance,
- `PERSON_GOAL_OBJECT_AT`($p2, obj, wp$): a goal from $p2$ where $obj$ and $wp$ are randomised object instances,

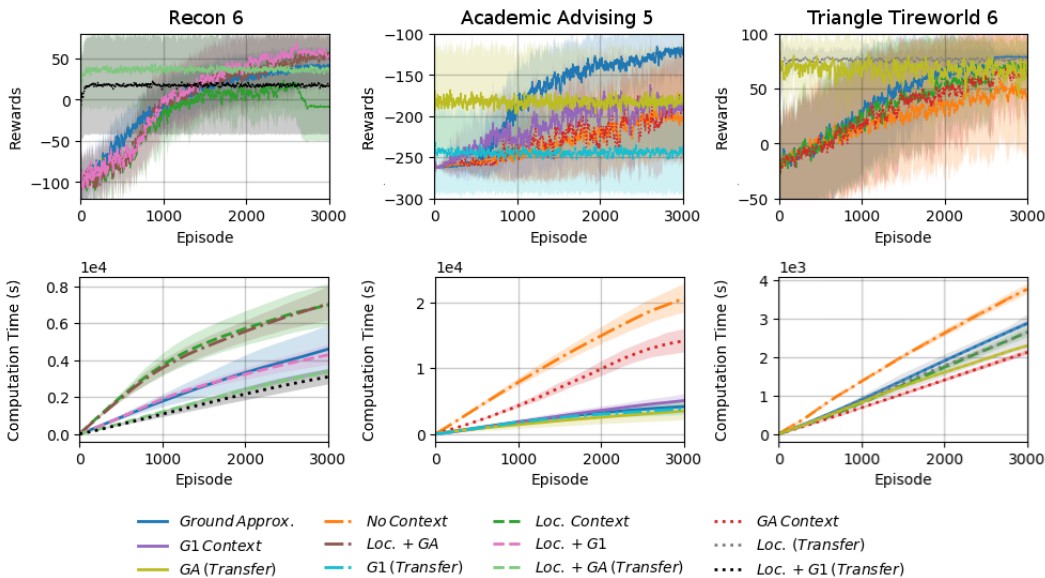

Figure 4: Results for an ablation study where different combinations of contextual grounding or no contextual grounding ($No\,Context$) are used. The performance for ground approximation ($Ground\,Approx.$) and transfer learning ($Transfer$) is also included. The results are aggregated over 10 randomised problems of three IPPC domains: `Recon`, `Academic Advising`, and `Triangle Tireworld`. The shading represents one standard deviation. Omitted experiments are either due to high computational costs or are inapplicable for a problem.

- PERSON_GOAL_OBJECT_AT($p2, obj, wp$): a goal from $p2$ where $obj$ and $wp$ are randomised object instances,

- PERSON_GOAL_OBJECT_WITH($p3, obj, p1$): a goal from $p3$ where $obj$ is a randomised object instance, and

- PERSON_GOAL_OBJECT_WITH($p3, obj, p2$): a goal from $p3$ where $obj$ is a randomised object instance.

The goals are randomised such that no $obj$ is involved in more than one goal. The goals from $p3$ involve another person (i.e., $p3$ requires the robot to bring $obj$ to the others). The results for SR3 are shown in Figure 3. The findings are similar to those discussed in Section 5 (see Figure 2): (1) the use of contextual grounding reduces computation time and improves performance, (2) transfer learning from a small scale problem (i.e., SR1) to a large scale problem (i.e., SR3) gives an initial performance boost, and (3) ground approximation does not scale well to large scale problems.

## A.6 Ablation Study for IPPC Domains

The IPPCs use several planning domains as benchmarks to evaluate the performance of competing planners. We conducted an ablation study on three benchmark domains: `Recon` (RC) [31], `Academic Advising` (AA) [33], and `Triangle Tireworld` (TT) [34]. These domains have problems which are numbered from 1 to 10 (a larger number represents a problem with a larger scale). We use Domain# to denote a problem numbered # for Domain. We conducted experiments for both small and large scale problems for each domain. A Q-function is learned in the small scale problems and transferred to the large scale problems. The results are shown in Figure 4. Since these domains do not involve durative actions, there are no results for mission time. The simulator used is RDDLSim.

In RC, an agent moves in a grid environment where there is a base, hazard, and objects. The goal is to use the tools to get readings on objects; tools can be damaged if the agent is at or adjacent to a hazard and this reduces the probability of getting a good reading. The agent can repair a tool at the base. We replaced the actions up, down, left, and right with move($wp$). This does not simplify

the problem but allows separate weights for each ground action of `move`. We used `RC3` and `RC6`, where the size of the state-action spaces are $2^{42} \times 28$ and $2^{55} \times 38$, respectively.

In `AA`, a student has to pass some required courses. The passing rate of a course $course_1$ depends on the number of prerequisites $course_2$ the student has passed. A reward is given at each time step for taking a course ($-1$) or retaking a failed course ($-3$), and if any required course has not been passed ($-5$). Passing a required course gives a reward of $5$. We used `AA3` and `AA5`, where the size of the state-action spaces are $2^{30} \times 16$ and $2^{40} \times 21$, respectively.

In `TT`, a vehicle moves in a grid environment to reach a goal location. There is a probability of $0.5$ of getting a flat tire when moving. The tire needs to be replaced with a spare tire; if there isn't one, a deadend is reached. The vehicle can load a spare tire if it doesn't have one and there is a spare at its current location. We used `TT3` and `TT6`, where the size of the state-action spaces are $2^{33} \times 242$ and $2^{59} \times 814$, respectively.

**Ablation Study.** We investigate the utility of contextual grounding by benchmarking the performance against two baselines: ground approximation and first-order approximation without contextual grounding. In `RC6`, first-order approximation outperforms ground approximation. The experiments for first-order approximation without contextual grounding could not be completed due to high computational costs. In `AA5`, first-order approximation performs poorly. This is because the goals could be dependent on other goals—the success of passing a course depends on the number of prerequisites passed. This cannot be represented by binary features. This issue is mitigated in a ground approximation due to the inclusion of every ground state fluent as features which serves as implicit counting. In `TT6`, the use of location context or goal context (they cannot be used together in this domain) improved the performance of first-order approximation which performed comparably to ground approximation.

Next, we consider the computational time. In `RC6`, location context and the combination of location and `GA` context have the highest computation time. `G1` considers only one goal at a time while `GA` considers every unachieved goal. The former has a significantly faster computation time. In `AA5`, the highest computation time is incurred when no context is used. Since contextual grounding reduces the set of possible substitutions for $\Phi$, the computation time decreases sharply. The same observation is seen in `TT6`.

**Knowledge Transfer.** We used a greedy policy generated by the Q-functions learned in the small scale problems (i.e., `RC3`, `AA3`, and `TT3`) to solve the large scale problems (i.e., `RC6`, `AA5`, and `TT6`). The Q-functions are kept unchanged while solving the large scale problems (i.e., knowledge transfer where no learning takes place). This is different from the setup in `SR` where an $\epsilon$-greedy policy is used and the Q-function continues to be updated (i.e., transfer learning). Figure 4 shows that in all problems, knowledge transfer performed significantly better than those which learn the Q-function online from scratch (i.e., online RL). Since the Q-function is not updated, the performance of knowledge transfer does not improve over episodes. Thus, in `RC6` and `AA5`, they are outperformed by online RL in later episodes. The computation time for knowledge transfer is lower than online RL as expected since no learning is done in the former.

