# OpenReview forum: "Generalised Task Planning with First-Order Function Approximation"
_robot-learning.org/CoRL/2021/Conference — CoRL2021 Poster_

### Official Review · Reviewer_jVmG · 2021-07-23

**Originality:** Good
**Technical Quality:** Good
**Clarity Of Presentation:** Poor
**Impact:** 2

**Recommendation:**

Weak Accept: I recommend accepting the paper, but will not argue for my recommendation if the majority of other reviewers have a different opinion.

**Summary:**

This paper focuses on how to generalize and transfer planning knowledge across a set of related tasks and domains. To this end, this paper presents a method for learning how to approximate q-values over first-order features (lifted features) for Relational Markov Decision Processes (RMDPs) and the ability to leverage contextual knowledge to reduce approximation errors in a specific task. By learning a generalized policy, it can be re-used and adapted for multiple planning problems to enable more efficient robot learning and planning. The method uses a feature discovery algorithm to identify first-order features so as to approximate first-order approximations of the Q-values for the RMDP, which enable the approximations to then be transferred to specific task instances. Feature discovery can also be used to discover more fine-grain features for a task so as to further reduce the approximation error based only on the first-order features.

**Issues:**

The authors should aim to make their technical writing more clear and grounded in the example, to provide the reader with a better understanding of how the first-order features are discovered, used for the first-order q-value approximations, and grounded for the instantiated planning problem.

**Reviewer Expertise:**

Poor: Limited knowledge of the area

**Strengths And Weaknesses:**

Strengths:
- This paper presents a novel approach to generalizing task planning by discovering first-order features that provide a feature space to learn lifted q-values and transfer the learned knowledge to specific task instances for robot planning.
- The preliminary section is succinct and straight-forward and easy to understand.
- The paper presents experiments for a service robot domain that is interesting and challenging since it involves waypoint locomotion, human interventions, and high-level object manipulation actions.
- The results of this paper demonstrate that the proposed method of learning over first-order features provides useful generalization across domains and reduces computational cost and memory usage.

Weaknesses:
- The technical section (section 4) is very challenging to read, and understanding the mechanics of the method is difficult due to the writing. The introduction of first-order features (section 4.1) and first-order approximations is not immediately clear, and the example provided (Example 1) is also challenging to read. The reviewer suggests starting with the example and focus on clearly presenting the domain and associated RMDP, rather than presenting all of details on first-order features first then describing the example 1. Because of this, it also challenging to follow the online feature discovery process (Section 4.2) and the contextual grounding of free variables (section 4.3). (The authors have addressed this weakness in the revision)


**Summary Of Recommendation:**

This paper presents a novel and interesting approach to learning generalized policies for transfer learning across planning domains, which is an important problem for robot learning. This paper also presents experimental results and analysis that support the claims that the new approach improves computational cost when learning in new domains. However, the clarity of the writing for the technical sections is not easy to follow and is unclear, making it challenging to fully assess the validity and impact of this paper.

---

> ### Author Response · Authors · 2021-08-26
> **Reply to Reviewer jVmG**
>
> We thank the reviewer for the feedback.
>
> > The technical section (section 4) is very challenging to read, and understanding the mechanics of the method is difficult due to the writing. The introduction of first-order features (section 4.1) and first-order approximations is not immediately clear, and the example provided (Example 1) is also challenging to read. The reviewer suggests starting with the example and focus on clearly presenting the domain and associated RMDP, rather than presenting all of details on first-order features first then describing the example 1. Because of this, it also challenging to follow the online feature discovery process (Section 4.2) and the contextual grounding of free variables (section 4.3).
>
> > The authors should aim to make their technical writing more clear and grounded in the example, to provide the reader with a better understanding of how the first-order features are discovered, used for the first-order q-value approximations, and grounded for the instantiated planning problem.
>
> We expanded on the introduction of Section 4. We explained the objective for our work and relate the various components of our methodology to our objective. We also improved the clarity of Sections 4.1, 4.2, and 4.3 and Examples 1, 2, and 3. The purpose of Example 1 is to give readers a concrete example of how features and free variables are represented in our work. Unfortunately, we can’t move Example 1 ahead in the text before the required notations and terminologies (e.g., free variables) are introduced.

---

> > ### Comment · Reviewer_jVmG · 2021-09-03
> > **Post-response**
> >
> > Thank you for addressing the confusion regarding the text, this has helped improve my understanding the of the paper. Due to my unfamiliarity with the area, I have only updated my suggestion to a weak accept.

---

> > > ### Author Response · Authors · 2021-09-04
> > > **Reply**
> > >
> > > We thank the reviewer for the improved score and the constructive comments.

---

### Official Review · Reviewer_uKVc · 2021-07-23

**Originality:** Good
**Technical Quality:** Very Good
**Clarity Of Presentation:** Good
**Impact:** 4

**Recommendation:**

Weak Accept: I recommend accepting the paper, but will not argue for my recommendation if the majority of other reviewers have a different opinion.

**Summary:**

The authors propose utilizing the first-order structure of some robotic domains to learn abstracted object oriented policies. The first-order structure refers to the fact that in some domains, multiple objects can belong to the same object type, and learning generalized policies over object and action types can lead to fast transfer on a new domain with same object types but different instantiations of the objects. The authors propose the use of an incremental feature learning algorithm (from prior work) along with contextual ground schemes (proposed by the authors).

The authors demonstrate the benefit of first-order learning by training in small service robot domain and successfully demonstrating transfer to a larger domain

**Issues:**

- There seems to be a lot more prior research on RL for relational MDPs [1],[2] and I found the related work section quite sparse and superficial. I would request the authors to develop the text there to indicate what is known and what is the key research gap that your work addresses.

[1] - Diuk, Carlos, Andre Cohen, and Michael L. Littman. "An object-oriented representation for efficient reinforcement learning." Proceedings of the 25th international conference on Machine learning. 2008.

[2] - Kim, Beomjoon, and Luke Shimanuki. "Learning value functions with relational state representations for guiding task-and-motion planning." Conference on Robot Learning. PMLR, 2020.

**Reviewer Expertise:**

Good: General knowledge of the area

**Strengths And Weaknesses:**

**Strengths:**

- The authors successfully demonstrated the utility of using a first-order abstraction to MDPs while learning the important conjunctive features from experience as well.

- The discussion in Section 4.4 about where the first order method is most applicable and useful is quite important, and I would like to see more of it in RL papers.

**Comments:**

- It seems that the grounding problem for the free variables could be a crucial bottleneck to successfully applying this approach broadly. The contextual grounding heuristics seem domain specific

- It would be interesting to see if the imposed relational structure shows any benefit in terms of faster learning curves for a non-relational MDP (i.e. without exploiting the first-order structure at all)

- As a non-expert in relational MDPs I found it hard to pinpoint the novel contribution of the paper. I believe the paper makes some important novel contributions, but the authors should provide an explicit description of their contributions vis-a-vis prior art in the introduction




**Summary Of Recommendation:**

I believe that the model-free problem formulation is novel and technically sound. I quite like the idea of using incrementally constructed logical features to decision-making. I would recommend acceptance

---

> ### Author Response · Authors · 2021-08-26
> **Reply to Reviewer uKVc**
>
> We thank the reviewer for the feedback.
>
> > It seems that the grounding problem for the free variables could be a crucial bottleneck to successfully applying this approach broadly. The contextual grounding heuristics seem domain specific
>
> The contextual knowledge can be consider domain-specific but is general and can be applied in every application where a mobile robot is tasked to perform some tasks (or goals). We require knowledge of the location of the mobile robot (for location context) and goals (for goal context) which are usually known in most planning problems and RL problems. Results for three benchmark domains from the International Probabilistic Planning Competition (IPPC) can be found in the supplementary materials. We refer the reviewer to Section B3 and Figure 2. These results demonstrate the generality of our method in non-robotic domains.
>
>
> > It would be interesting to see if the imposed relational structure shows any benefit in terms of faster learning curves for a non-relational MDP (i.e. without exploiting the first-order structure at all)
>
> We thank the reviewer for the suggestion. In a non-relational MDP where the planning problem does not have a first-order structure, our assumption is violated. Nevertheless, it is interesting to see if our work would still accelerate learning in non-relational MDPs via transfer learning. We are exploring other means of transfer learning other than the transfer of the Q-function approximation and this suggestion will fit in well with our ongoing work.
>
>
> > As a non-expert in relational MDPs I found it hard to pinpoint the novel contribution of the paper. I believe the paper makes some important novel contributions, but the authors should provide an explicit description of their contributions vis-a-vis prior art in the introduction
>
> We have strengthened the statement of our contributions in the Introduction. We reproduce the writings here for ease of reference.
> > Our contribution is threefold. First, we propose a method to generate a first-order feature space automatically given a RMDP without the use of a model, expert knowledge, or training data. Second, we implement a online, model-free RRL method which learns a first-order linear function approximation of the Q-function given an initial set of first-order features. The approximated Q-function induces a generalised policy which allows transfer learning between related problems independent of the objects, number of objects, initial states, and goal states. Third, we introduce the concept of contextual knowledge to reduce the granularity in the first-order function approximation, improve plan optimality, and reduce computational cost. We evaluate our method empirically on randomised problems of a service robot domain and in different simulation environments.
>
>
> > There seems to be a lot more prior research on RL for relational MDPs [1],[2] and I found the related work section quite sparse and superficial. I would request the authors to develop the text there to indicate what is known and what is the key research gap that your work addresses
>
> We thank the reviewer for the suggested readings. In our amended paper, we have included more prior works and strengthen the discussion on the connections between our work and prior work. This can be found in Section 2. The key research gap we seek to address is a model-free RRL algorithm which is domain-independent and only requires trivial human knowledge (for contextual grounding). RRL allows transfer learning between problems with first-order structure. The key differences between prior works on RRL and our work is that we utilise conjunctive first-order features with free variables which has a richer representation of the Q-function. Related works use first-order decision trees and ignore free variables. This limits their approaches to problems with simple goals (e.g., stack block A on block B). If we prune features with free variables, we would have lost the expressive representation of our first-order function approximation. Therefore, we believe our method can solve a larger class of problems involving more complex goals (e.g., put item A on table B AND bring item C to person P1) than prior works. Furthermore, it was noted in [Driessens & Ramon, 2003] that the performance of decision trees relied on the correct split of nodes. Thus, it might not be suited for online RL. We are unable to test these hypotheses in our paper due to the different language representations used in our work (RDDL) and in their work (datalog). Please refer to our reply to reviewer dCnP on the challenges of benchmarking with other works.
>
>
> - Kurt Driessens and Jan Ramon. Relational instance based regression for relational reinforcement 421 learning. In Proc. ICML, pages 123–130, 2003.

---

### Official Review · Reviewer_dCnP · 2021-07-25

**Originality:** Good
**Technical Quality:** Good
**Clarity Of Presentation:** Good
**Impact:** 2

**Recommendation:**

Weak Accept: I recommend accepting the paper, but will not argue for my recommendation if the majority of other reviewers have a different opinion.

**Summary:**

The paper is focused on generalization over different planning problems. Their core idea is to exploit the first-order structure of the robot-environment interaction to provide an approximation of the Q-function that can generalize to large-scale problems and adapt to different environments. This approach is useful in situations where a model is not available and where the number of interactions with the environment is restricted, which is the case for real world scenarios.
Their approach can deal with different objects, different number of objects, different initial conditions and final goals, without having to learn a different policy for every situation.
In practice, they use model-free relational RL with online feature discovery using another algorithm from literature, iFDD+.
They develop their method in the context of Relational MDP (RMDP), that allows them to learn a generalized policy $\pi$ able to solve different planning problems (any of the ground MDPs of the RMDP). The policy is induced by a linear approximation of the Q-function, defined over features discovered online by iFDD+.

**Issues:**

- I find the writing style to be a bit raw and overloaded with notation (e.g. in Example 2 and 3). I appreciate that you provided said examples to make the reader understand how free variables are grounded. But maybe they could be complemented by another set of figures/a small scheme, also considering that there are only 2 figures in the paper.
- The reader does not get an idea of how your work compares with related work, baselines are missing

**Reviewer Expertise:**

Fair: Some knowledge of the area

**Strengths And Weaknesses:**

### Strengths
The authors focus on a problem that is not very popular in the community, yet quite relevant. They provide a simple way to derive a generalized policy, based on a first-order approximation making the formulation independent of state, action and object space. This is important for transfer learning, as showed in the experiments.


### Weaknesses
Nevertheless, the authors do not present any baseline related to existing literature work on transfer learning. In addition to this, only one type of domain/task is considered in the main paper, the service robot domain illustrated in figure 1.

**Summary Of Recommendation:**

The idea of the paper is to use domain knowledge to do contextual grounding and then use this to train a linear Q-function. The features are expressed as a set of conjunctive state predicates, discovered with the use of an off the-shelf algorithm (iFDD+).
The main reason motivating my recommendation us that there is not a single baseline comparison with related work on transfer learning, nor a comparison with any other Q-learning method from scratch. Like this the reader cannot understand how it compares with previous work or how faster it is compared to a setting without transfer, for example vs a policy learned with any SOTA model-free RL.

---

> ### Author Response · Authors · 2021-08-26
> **Reply to Reviewer dCnP**
>
> We thank the reviewer for the feedback.
>
> > In addition to this, only one type of domain/task is considered in the main paper, the service robot domain illustrated in figure 1
>
> We presented results for three benchmark domains from the International Probabilistic Planning Competition (IPPC) in our supplementary materials. We refer the reviewer to Section B3 and Figure 2. The purpose of these results is twofold: (1) to demonstrate our work is domain-independent, and (2) to evaluate our work on domains which are not handcrafted by us. Due to space limitations and because CORL is a robotics conference, we decided that the results for the IPPC domains might be of less interest to the readers.
>
>
> > Nevertheless, the authors do not present any baseline related to existing literature work on transfer learning.
>
> > The main reason motivating my recommendation us that there is not a single baseline
> comparison with related work on transfer learning, nor a comparison with any other Q-learning method from scratch. Like this the reader cannot understand how it compares with previous work or how faster it is compared to a setting without transfer, for example vs a policy learned with any SOTA model-free RL.
>
> > The reader does not get an idea of how your work compares with related work, baselines are missing
>
> We understand the difficulty in evaluating the impact of the contribution of our work without any baseline comparisons. However, this has often been the case in the area of transfer learning for reinforcement learning (TLRL) as pointed out by [Taylor & Stone, 2009]*.
>
> The difficulty in benchmarking TLRL algorithms is due to several reasons. Different works:
> 1. utilise different assumptions (e.g., allowed dissimilarity in tasks),
> 2. transfer different forms of knowledge (e.g., policies, models, options, Q-functions),
> 3. address different classes of problems (e.g., discrete domains, continuous domains, relational domains), and
> 4. allow different learners (e.g., TD learning, linear programming, policy search).
>
> To the best of our knowledge,  [Taylor & Stone, 2009] is one of the most recent survey papers for TLRL. Another survey paper, [Torrey & Shavlik, 2010], did not mention any benchmark and did not compare different works; instead, they evaluated each work on its own merit. We have cited all significant, related works for TLRL (excluding some papers which are essentially incremental variants of these works) in our amended paper and none of them have performed comparisons with a baseline from the works of another author.
>
> Lastly, we have indeed compared our work with a “Q-learning method from scratch” which is labelled ‘Ground’ in Figure 2. This uses the ground approximation (i.e., the ‘vanilla’ linear function approximation setup). We used Double Q-learning [Hasselt, Guez, & Silver, 2016], which is a state-of-the-art model-free TD learning algorithm, for all our experiments.
>
> *On a side note, recent survey papers on transfer learning for classification, regression, and clustering [Pan & Yang, 2010; Weiss, Khoshgoftaar, & Wang, 2016; Zhuang, et al., 2020] compared different transfer learning methods. We emphasize that classification, regression, and clustering are very different from reinforcement learning and these subfields of AI have different practices (e.g., standardised benchmarks, open source libraries, publicly available codes, etc.).
>
> - Matthew E. Taylor, and Peter Stone. "Transfer learning for reinforcement learning domains: A survey." Journal of Machine Learning Research 10, no. 7 (2009).
> - Lisa Torrey, and Jude Shavlik. "Transfer learning." In Handbook of Research on Machine Learning Applications and Trends: Algorithms, Methods, and Techniques, pp. 242-264. IGI Global, 2010.
> - Hado van Hasselt, Arthur Guez, David Silver. “Deep reinforcement learning with double Q-learning.” AAAI, pp. 2094-2100, 2016.
> - Sinno Jialin Pan, and Qiang Yang. "A survey on transfer learning." IEEE Transactions on knowledge and data engineering 22, no. 10 (2009): 1345-1359.
> - Karl Weiss, Taghi M. Khoshgoftaar, and DingDing Wang. "A survey of transfer learning." Journal of Big Data, pp. 1-40, 2016.
> - Fuzhen Zhuang, Zhiyuan Qi, Keyu Duan, Dongbo Xi, Yongchun Zhu, Hengshu Zhu, Hui Xiong, and Qing He. "A comprehensive survey on transfer learning." Proceedings of the IEEE 109, pp. 43-76, 2020.
>
>
> > I find the writing style to be a bit raw and overloaded with notation (e.g. in Example 2 and 3). I appreciate that you provided said examples to make the reader understand how free variables are grounded. But maybe they could be complemented by another set of figures/a small scheme, also considering that there are only 2 figures in the paper.”
>
> We thank the reviewer for the suggestion. We have added more description to Examples 2 and 3 and removed some notations from the latter. We have also updated Figure 2 to illustrate location context and goal context.

---

> > ### Comment · Reviewer_dCnP · 2021-09-05
> > **Response to authors**
> >
> > I thank the authors for their response. I now understand that a thorough comparison with existing methods is quite an endeavor considering all the different aspects coming into play in TLRL.
> > I have read the additions to the examples and the sections, however, I believe the paper is still hard to parse after a first reading. Given that there are no baseline comparisons, clarity of exposure thus becomes a fundamental concern.
> > Nonetheless, the contribution of this paper could have some impact on the field and be very useful to practitioners. For this reason, I will increase my score to Weak Accept.

---

> > > ### Author Response · Authors · 2021-09-06
> > > **Response to Reviewer dCnP**
> > >
> > > We thank the reviewer for the feedback and improved score. Our baseline is labelled "Ground" in Figure 2. This is the configuration which uses every ground state predicates as the set of initial features (i.e., our proposed method is not in play here). Unfortunately, our service robot domain has too many state predicates and running this baseline is intractable in our larger scale problems. We did show the baseline in other benchmark domains in our supplementary materials. We can place these supplementary results in the appendix of the paper.

---

### Official Review · Reviewer_rhBR · 2021-07-26

**Originality:** Good
**Technical Quality:** Good
**Clarity Of Presentation:** Fair
**Impact:** 2

**Recommendation:**

Weak Reject: I recommend rejecting the paper, but will not argue for my recommendation if the majority of other reviewers have a different opinion.

**Summary:**

The paper considers task planning in relational domains using relational RL based on learned Q functions.
One challenge in this setup is representing the space of all possible states compactly.
To address this, the authors propose to define the Q function as a linear combination of lifted features. This allows those features to be queried for different MDPs, even with different numbers of objects.
In order to evaluate the Q function, however, these features must be fully grounded. This is achieved by introducing a context, which is a reduced set of objects from which the features can be grounded.
The work largely builds upon iFDD+ [1] with the difference of how the features are defined.


**Issues:**

- The memory limit seems rather arbitrary and outdated. 4gb of memory is not much in today's standards.
- Notation: Q(s,a) is a value, not a function. The function is called Q. Similarly for \phi(s,a).
- Section 4.1 could be written more clearly.
- I don't particularly like that the formal method is explained using examples only, especially in sec 4.3

**Reviewer Expertise:**

Fair: Some knowledge of the area

**Strengths And Weaknesses:**

Strength:
- The idea of representing lifted features instead of grounded ones is compelling

Weaknesses:
- However, it is unclear to me if the idea of lifted features really works well when only considering task planning domains without geometric conditioning on the features. Can it be shown that the lifted approximation really is a valid assumption? In the case of Loc + goal context, how many free variables are left?
- For a pure task planning scenario, the number of variables does not seem to be particularly high. I would assume that if the number of objects is larger, then the lifted approximation becomes less and less reliable.

**Summary Of Recommendation:**

As such, the paper is sound, but the method does not solve a particularly challenging or urgent topic in robotics, therefore, the potential impact is limited.

---

> ### Author Response · Authors · 2021-08-26
> **Reply to Reviewer rhBR (Part 1/2)**
>
> We thank the reviewer for the feedback.
>
> > However, it is unclear to me if the idea of lifted features really works well when only considering task planning domains without geometric conditioning on the features.
>
> Our method is domain-independent and agnostic to the features used (i.e., does not need handcrafted features); it can be applied to any task-level problem with first-order structure. It does not deal directly with geometric properties, instead, it learns and reasons at the task-level. We demonstrated our method is domain-independent with experimental results on three benchmark domains presented in the supplementary materials. In these non-robotic domains, there is no planning in the geometric space (e.g., motion planning).
>
> In our Service Robot domain, planning in the geometric space is dealt by ROS packages. We implemented ROS packages for every task-level action. For example, when find_person(P1) is executed, the robot moves along an exploration path to search for person P1. We used ROS packages from publicly available libraries for collision avoidance, path planning, etc. Our method does not require geometric information (e.g., robot is at X=1.5, Y=3.2) from the simulator (or real robot); it only requires symbolic information (e.g., robot_at(WP1)).
>
>
> > Can it be shown that the lifted approximation really is a valid assumption?
>
> Our first-order approximation works well under the following circumstances:
> 1. Planning problem has a first-order structure
> 2. Goal-directing policy can be induced by a linear approximation of the Q-function
> For (1), this is often the case in many robotic applications. A robot interacting with its environment is certain to affect the environment in the same way regardless of which object it interacts with (e.g., picking up an item, opening a door). For (2), our first-order approximation is limited to reason about at most two objects for each object type (one for the bound variable and one for the free variable). Thus, in some domains (such as Academic Advising in the supplementary materials) where this is violated, the performance is reduced.
>
>
> > In the case of Loc + goal context, how many free variables are left?
>
> There is one free variable of type ‘person’, *person, remaining after applying Loc + Goal context if the goal does not involve a ‘person’. There are four object types in our service robot domain: ‘item’, ‘person’, ‘robot’, ‘waypoint’. *item is grounded with goal context, *person is grounded with goal context if the goal is to deliver an item to a person, there is no free variables of type ‘robot’ since there is only one robot, and *waypoint is grounded with location context. If the goal does not involve a ‘person’ (i.e., bring an item to a location rather than to a person), then goal context will not ground the *person. However, under this circumstance, the grounding of *person is inconsequential since the policy to achieve the goal does not involve a person.
>
>
>
> > The memory limit seems rather arbitrary and outdated. 4gb of memory is not much in today's standards.
>
> We ran our experiments on a computer cluster which imposes a memory limit for each job (experiment) to 4 GB. This limit is imposed on only the RRL algorithm and not on the ROS simulator which runs on a laptop with 16 GB of RAM. We have now clarified this point in our paper. We agree that 4 GB is not much, but as mentioned, this is only for the RRL algorithm. In many robots, there are other computational processes running in parallel (e.g., computer vision, SLAM, planning, etc.). Thus, we think that 4 GB is a reasonable limit and is comparable to other works such as [Sanner & Boutilier, 2009] which uses 2 GB.
>
> - Sanner, Scott, and Craig Boutilier. "Practical solution techniques for first-order MDPs." Artificial Intelligence 173, no. 5-6 (2009): 748-788.

---

> > ### Author Response · Authors · 2021-08-26
> > **Reply to Reviewer rhBR (Part 2/2)**
> >
> >
> > > For a pure task planning scenario, the number of variables does not seem to be particularly high. I would assume that if the number of objects is larger, then the lifted approximation becomes less and less reliable.
> >
> > Indeed, as the number of objects increase, the number of possible substitutions for free variables increase proportionally. This actually strengthens the utility of our method of using contextual knowledge to reduce the number of possible substitutions. We see our concept of contextual knowledge as a first step towards more general use of first-order approximation where other forms of contextual knowledge can be applied in addition to the ones we introduced in the paper. For future work, we can consider subgoals or landmarks [Pereira, Oren, & Meneguzzi, 2017] (complementing our goal context), or other means to learn the contextual grounding given a collection of observations.
> >
> > - Ramon Pereira, Nir Oren, and Felipe Meneguzzi. "Landmark-based heuristics for goal recognition." Proceedings of the AAAI Conference on Artificial Intelligence. Vol. 31. No. 1. 2017.
> >
> >
> > > As such, the paper is sound, but the method does not solve a particularly challenging or urgent topic in robotics, therefore, the potential impact is limited.
> >
> > We disagree with this view. Our method allows generalisation and transfer learning (e.g., Sim2Real) for task planning in robotics and does not require a model. This suits robotic applications where the model is too complex to be handcoded and training data is expensive to collect. We believe that Sim2Real is an existing challenge of high interest as it is still being addressed actively by the research community. We would appreciate if the reviewer would elaborate on this view to promote further discussion.
> >
> >
> >
> > > Section 4.1 could be written more clearly.
> >
> > We appreciate the comment and have improved the presentation of Section 4.1.
> >
> >
> > > I don't particularly like that the formal method is explained using examples only, especially in sec 4.3
> >
> > Due to space limitations, we omit details on our method and used examples to give a more intuitive understanding of the semantics of our approach. We have provided details of our method in the supplementary materials (Sections A.1 and A.2, Algorithms 1, 2, and 3). In our amended paper, we have included some formal noations in Section 4.3.

---

> > > ### Comment · Reviewer_rhBR · 2021-09-04
> > > **Re: Official Review of Paper127 by Reviewer rhBR**
> > >
> > > I thank the authors for their detailed response.
> > >
> > > >A robot interacting with its environment is certain to affect the environment in the same way regardless of which object it interacts with (e.g., picking up an item, opening a door).
> > >
> > > This is one of the main reasons why I have difficulties judging the impact of the paper.
> > > This assumption is more or less always violated, if one does not constrain to simple scenarios or a lot of hand-engineered motion primitives.
> > >
> > > Nevertheless, I won't argue against publication, if the other reviewers have a different opinion on this point.

---

> > > > ### Author Response · Authors · 2021-09-04
> > > > **Comment on Assumption**
> > > >
> > > > > This assumption is more or less always violated, if one does not constrain to simple scenarios or a lot of hand-engineered motion primitives.
> > > >
> > > > We would like to clarify that our statement applies in the context of our work which deals with task planning and not motion primitives. That is, our state space is represented symbolically with state predicates. For example, a robot opens door A at location WP1 and this changes the state predicate door_is_open(A) from a value of false to true. This transition applies even if it is door B, C, ..., at any location. We describe a deterministic transition but the same applies to a probabilistic transition and all of our domains (including those in supplementary materials) are probabilistic. Thus, our work is not constraint to simple scenarios and do not use hand-engineered features. We use off-the-shelf motion planning ROS packages which deals with reasoning in the geometric space. Our work do not deal directly with these; rather, our approach decouples planning at the task level and at the geometric space level.

---

### Meta-Review · Area_Chair_o325 · 2021-08-10

**Recommendation:** Accept (Poster)
**Confidence:** 4

**Metareview:**

To enable generalization across different instances of a problem, the authors propose to learn a Q-function (and by extension a policy) defined over a more abstract state-action space that is oblivious to most of the specifics of the different entities composing the state save for their type (type such as 'person', 'object', 'location'...). The main technical contribution of the paper is how to resolve the ambiguity of grounding the Q-function to a specific problem instance if there are several entities of the same type.

Pros:
- The reviewers find the idea compelling, the setting of relational RL original and relevant for transfer and planning.
- The consensus among the reviewers is that the experiments successfully demonstrate the ability of the proposed algorithm to generalize across different instances.

Cons:
- There are concerns regarding the generality of the proposed contextual grounding method, which is the main contribution of the paper. R-r (Reviewer rhBR) is unclear how a lifted planner could work without geometric information, and deplores that the grounding is only described informally through examples. R-u similarly thinks that the grounding appears heuristic and domain specific.
- There are issues with the clarity of the presentation (R-r, R-j, R-d), especially the technical sections.
- R-d notes that comparisons to baselines are missing and that it is in general hard to appreciate the work in the context of the existing literature. R-u finds the related work section superficial and proposes some additional references to enrich it and better highlight the contributions of the paper.

The general consensus among reviewers is leaning towards rejection, but the authors are highly encouraged to engage with the reviewers and revise their manuscript accordingly. Among other things, addressing concerns regarding the generality of the method by improving the clarity of the technical section and better highlighting the contributions would greatly improve the quality and potential impact of the paper.

I thank the authors for engaging with the reviewers and clarifying their submission. The reviewers are more positive about this work after the rebuttal period. The submission is original and improving generalization of RL through added abstraction is a commendable direction. I thus recommend the paper for presentation.

---

### Decision · Program_Chairs · 2021-09-13

**Decision:**

Accept (Poster)

**Comment:**

To enable generalization across different instances of a problem, the authors propose to learn a Q-function (and by extension a policy) defined over a more abstract state-action space that is oblivious to most of the specifics of the different entities composing the state save for their type (type such as 'person', 'object', 'location'...). The main technical contribution of the paper is how to resolve the ambiguity of grounding the Q-function to a specific problem instance if there are several entities of the same type.

Pros:
- The reviewers find the idea compelling, the setting of relational RL original and relevant for transfer and planning.
- The consensus among the reviewers is that the experiments successfully demonstrate the ability of the proposed algorithm to generalize across different instances.

Cons:
- There are concerns regarding the generality of the proposed contextual grounding method, which is the main contribution of the paper. R-r (Reviewer rhBR) is unclear how a lifted planner could work without geometric information, and deplores that the grounding is only described informally through examples. R-u similarly thinks that the grounding appears heuristic and domain specific.
- There are issues with the clarity of the presentation (R-r, R-j, R-d), especially the technical sections.
- R-d notes that comparisons to baselines are missing and that it is in general hard to appreciate the work in the context of the existing literature. R-u finds the related work section superficial and proposes some additional references to enrich it and better highlight the contributions of the paper.

The general consensus among reviewers is leaning towards rejection, but the authors are highly encouraged to engage with the reviewers and revise their manuscript accordingly. Among other things, addressing concerns regarding the generality of the method by improving the clarity of the technical section and better highlighting the contributions would greatly improve the quality and potential impact of the paper.

I thank the authors for engaging with the reviewers and clarifying their submission. The reviewers are more positive about this work after the rebuttal period. The submission is original and improving generalization of RL through added abstraction is a commendable direction. I thus recommend the paper for presentation.